# Watching subtitled videos with the sound off affects viewers' comprehension, cognitive load, immersion, enjoyment, and gaze patterns: A mixed-methods eye-tracking study

Agnieszka Szarkowska[1]*, Valentina Ragni[1], Sonia Szkriba[1], Sharon Black[2], David Orrego-Carmona[3,4], Jan-Louis Kruger[5,6]

1 Faculty of Applied Linguistics, University of Warsaw, Warsaw, Poland, 2 School of Politics, Philosophy, Language and Communication Studies, University of East Anglia, Norwich, United Kingdom, 3 School of Modern Languages and Cultures, University of Warwick, Coventry, United Kingdom, 4 Department of Linguistics and Language Practice, University of the Free State, Bloemfontein, South Africa, 5 Department of Linguistics, Macquarie University, Sydney, Australia, 6 UPSET Focus Area, North-West University, Vanderbijlpark, South Africa

* a.szarkowska@uw.edu.pl

## Abstract

Every day, millions of viewers worldwide engage with subtitled content, and an increasing number choose to watch without sound. In this mixed-methods study, we examine the impact of sound presence or absence on the viewing experience of both first-language (L1) and second-language (L2) viewers when they watch subtitled videos. We explore this novel phenomenon through comprehension and recall post-tests, self-reported cognitive load, immersion, and enjoyment measures, as well as gaze pattern analysis using eye tracking. We also investigate viewers' motivations for opting for audiovisual content without sound and explore how the absence of sound impacts their viewing experience, using in-depth, semi-structured interviews. Our goal is to ascertain whether these effects are consistent among L2 and L1 speakers from different language varieties. To achieve this, we tested L1-British English, L1-Australian English and L2-English (L1-Polish) language speakers (n = 168) while they watched English-language audiovisual material with English subtitles with and without sound. The findings show that when watching videos without sound, viewers experienced increased cognitive load, along with reduced comprehension, immersion and overall enjoyment. Examination of participants' gaze revealed that the absence of sound significantly affected the viewing experience, increasing the need for subtitles and thus increasing the viewers' propensity to process them more thoroughly. The absence of sound emerged as a global constraint that made reading more effortful. Triangulating data from multiple sources made it possible to tap into some of the metacognitive strategies employed by viewers to maintain comprehension in the absence of sound. We discuss the implications within the context of the growing trend of watching subtitled videos without sound, emphasising its potential impact on cognitive processes and the viewing experience.

**Data Availability Statement:** All relevant data are available here: https://osf.io/9nmfx/.

**Funding:** This study was conducted within the WATCH-ME project funded by the National Science Centre OPUS 19 programme 2020/37/B/HS2/00304. The principal investigator is Agnieszka Szarkowska, whereas Valentina Ragni is employed as a post-doc and Sonia Szkriba as a PhD candidate researcher in the project.

**Competing interests:** The authors have declared that no competing interests exist.

## Introduction

Millions of people worldwide now regularly watch video content with subtitles. The growing demand for subtitles can be attributed at least in part to the widespread availability of streaming platforms such as Netflix, Hulu, Amazon Prime, or Disney+, the large volume of video content on social media platforms, as well as the implementation of accessibility laws like the Americans with Disabilities Act and the EU Audiovisual Media Services Directive. Subtitles offer benefits not only to individuals who are deaf or hard of hearing, by enabling them to access the audio content, but also to wider audiences, for example by helping them understand foreign content or unfamiliar accents [1, 2]. Recent research indicates that it is younger generations, particularly Generation Z and millennials, who increasingly choose to enable subtitles, even in countries like the USA or Poland, where subtitling has not been prevalent traditionally [3–7]. Reasons cited for watching content with subtitles include unclear audio, fast speech rates, background music, challenging accents, the desire to focus, language learning, and not wanting to disturb family members or roommates [8–10].

The rise in subtitled content corresponds with changes in viewer behaviour. One notable trend is the practice of watching videos with muted sound on mobile devices, particularly short-form content on social media, in various settings like public places and during travel. A study conducted by Verizon Media and Publicis Media found that up to 69% of consumers prefer watching videos without sound in public spaces, with 25% choosing this approach in private settings [11]. In such cases, viewers largely depend on subtitles for content comprehension.

From a film studies perspective, sound serves multiple important functions in a film: it ensures narrative continuity and coherence, creates mood, and enhances authenticity [12, 13]. The presence of sound significantly shapes the making-meaning process and content engagement [14]. When viewers choose to switch off the sound, they are deprived of three vital elements: dialogues (including prosody), sound effects, and music. Despite the critical role sound plays in enhancing the cinematic experience, as reflected in prestigious awards such as the Academy Award for Best Sound and Best Original Score, and substantial investments in advanced film sound technologies, it is surprising that many viewers opt to watch audiovisual content without sound.

In this study, we examine this new trend of watching audiovisual content with no sound and investigate its impact on viewers' comprehension, recall, cognitive load, immersion, and gaze patterns. By adopting a mixed methods approach [15] and analysing data from self-reports, eye movements, comprehension and recall post-tests, and semi-structured interviews, we seek to gain insights into why people choose to watch audiovisual content without sound and how this decision impacts their overall viewing experience.

### Theoretical framework

Watching subtitled video requires the coordination of complex mental processes that have not been explored extensively in the literature. There are some theoretical models and frameworks to explain the integration of text, image and sound, such as the integrated model of text and picture comprehension [16]; the multimodal integrated-language framework [17, 18]; and the cognitive theory of multimedia learning [19, 20]. The framework by Liao at al. [17] investigates this integration in the context of dynamic texts like video with subtitles and a soundtrack. Both the model by Schnotz [16] and Mayer's theory [20] understand the cognitive architecture as based on input from auditory and visual sensory modalities and a limited working memory capacity. They highlight how split attention in contexts like watching subtitled video without sound could impede comprehension. In Schnotz's model [16], split attention would result in

both descriptive processing (of words) and depictive processing (of sounds and images) being dependent on visual perception, thereby increasing cognitive load. In Mayer's theory [20], the multimedia principle holds that a combination of words and pictures are better for learning than words alone, whereas the modality principle assumes that presenting information in two modalities is better for comprehension than presenting information only in one modality. This theory, like Schnotz's model, would imply that subtitled video without sound would therefore increase cognitive load.

Neither Schnotz's model nor Mayer's theory, however, specify how the processing of one source of information might impact on another (e.g., video and subtitles). The multimodal integrated-language framework of Liao et al. [17, 18] builds on models of eye movement control during reading to predict how the reading of subtitles might be impacted by the concurrent presence of visual and auditory information. The framework assumes that reading is a serial process (i.e., sentence processing is contingent on the strictly sequential identification of words) and that objects in a scene likewise have to be fixated sequentially for identification due to the limitations of visual perception. However, the auditory processing of speech and sounds and the tracking of previously identified objects can occur in parallel with these serial processes. In the absence of sound, the benefit of parallel processing of sound disappears, forcing the viewer to rely on the sequential processing of image and subtitles.

This article will attempt to confirm some of the findings of Liao et al. [17], by exploring whether greater reliance on subtitles in the absence of sound will result in more time on the subtitles, longer fixations, and shorter saccades (or lower word skipping), reflecting the prioritization of the subtitles when the benefit of parallel processing of sound is absent.

## Cognitive load

The theoretical frameworks above are based on the fact that the components of the human cognitive architecture have limited capacity. According to cognitive load theory [21], limited working memory capacity might prevent novel information from entering long-term memory, which is why new information should be introduced in a way that does not create needless working memory load. In line with the theories discussed above, removing the sound from subtitled film would force viewers to process all information through the visual channel sequentially, without the benefit of concurrent processing of sound, thereby increasing cognitive load. A previous study by Łuczak [22] in the context of subtitled video showed that when watching subtitled videos without sound, viewers reported higher cognitive load compared to the condition with the sound. Interestingly, despite higher cognitive load and contrary to what one would expect, viewers' comprehension was not affected by taking away the sound. A similar result was reported by Liao et al. [17], who found no significant differences in comprehension between subtitled videos with and without audio. Overall, this could mean that although viewers may need to expend more cognitive resources when watching videos without sound, this extra effort compensates for any potential loss in comprehension.

## Immersion and enjoyment

Filmmakers dedicate considerable effort to draw viewers in the narrative world and evoke their suspension of disbelief [23]. The suspension of disbelief involves viewers seemingly disregarding "reality to enter into and become engaged with the circumstances and the inhabitants of fictional worlds" [24]. Immersion, often described as the sensation of "being there" [25] and "becoming lost in a fictional reality" [26], is a multi-faceted concept originating from studies in virtual reality, encompassing various dimensions and experiences [27]. It can be approached

from different angles and has been referred to using various terms in literature, including transportation, engagement, flow, presence, involvement, or absorption [27–30].

Previous research has revealed several factors contributing to higher immersion [31]. It is generally believed that "the greater the number of human senses for which a medium provides stimulation (i.e., media sensory outputs), the greater the capability of the medium to produce a sense of presence" [32]. Therefore–and importantly for this study–the presence of both the visual and auditory layers could trigger higher immersion than media without sound, as sound effects, prosody and music play a significant role in enhancing immersion and the overall film-watching experience by evoking emotions, and by supporting storytelling [28] and language processing. Yet, we still know very little about how the complete removal of the soundtrack affects immersion, which is a question we address in this study.

Viewers also seek to derive enjoyment from watching audiovisual content. Enjoyment is often defined as "a pleasurable response to entertainment media" [33]. Viewers typically experience the feeling of enjoyment as a result of "emotional affiliations with characters and the outcomes associated with those characters as presented in the narrative" [34]. If sound–including the cadence, tone, pitch, prosody and other paralinguistic parameters of a character's speech–is taken away, viewers' emotional reactions towards characters may be negatively affected, resulting in lower enjoyment of the audiovisual media entertainment content overall [34].

### Evidence from eye-tracking research

Previous eye-tracking research on subtitle reading has consistently demonstrated that subtitles are great gaze attractors [35, 36]. Viewers tend to look at subtitles when they are displayed and this phenomenon holds true regardless of viewers' prior experience with subtitling or the type of subtitles (interlingual, intralingual, or bilingual). Additionally, access to the film soundtrack–understood as either the presence/absence of sound or familiarity with the language of the soundtrack–is known to influence how viewers engage with subtitled content [36–40].

Where sound is absent, it is reasonable to expect that viewers will rely more on the subtitles, as they constitute the main source of verbal information. However, eye-tracking studies to date provide little and conflicting evidence regarding the amount of time viewers spend reading subtitles with and without sound [36, 38, 41]. The implication of spending more time in the subtitle area is that it comes at the cost of having less time to follow the unfolding on-screen action, which in turn could have a negative impact on understanding the plot, immersing in the story world, picking up on subtle visual cues, and enjoying the filmic product. Unfortunately, previous relevant research has not tested comprehension, immersion, enjoyment, and cognitive load together, so it is difficult to establish whether this increased time spent reading the subtitles without sound has a detrimental effect on the viewing experience. In this study, we therefore combine eye-tracking with measures of comprehension, immersion, enjoyment and cognitive load precisely to obtain a fuller picture of whether and how the absence of sound affects viewers' overall engagement and the viewing experience as a whole.

Previous eye-tracking research on subtitle reading has found that when viewers watch subtitled videos without sound, they tend to display shorter progressive saccades, less subtitle skipping, and higher fixation counts [17, 22], suggesting that removing the soundtrack has a discernible effect on reading patterns. Interestingly, however, the absence of sound does not seem to affect mean fixation duration (MFD), usually an indicator of deeper or more demanding cognitive processing [42]. It is therefore possible that, although most research to date shows that sound alters the reading experience, it may not impinge on it negatively in terms of cognitive effort (as evidenced by MFD), still allowing viewers to extract the information they

need to fully comprehend the content, follow the storyline, and enjoy the experience. Although Liao et al. [17] and Łuczak [22] did not assess enjoyment or immersion, they did assess comprehension, and indeed did not find significant differences between the sound-on and sound-off conditions. In the present study we examine eye movement measures together with post-tests, self-reports and interviews to obtain a more nuanced understanding of viewers' perception, comprehension and reading patterns of both L1 and L2 subtitles. Below, we present a more detailed review of relevant subtitle research involving L2 populations.

## The impact of sound on L1 and L2 viewers

To date, research investigating multimodal processing where multiple sources of information are available (sound, images, subtitles) has focussed on testing whether subtitles are useful, particularly for comprehension and vocabulary learning (for a meta-analysis, see [43]; for a recent review, see [44]). Research has amply confirmed that adding same-language (i.e., intralingual) subtitles to foreign-language videos enhances comprehension compared to listening-whilst-watching without subtitle support [45–47]. In comparison, other aspects of the multimodal viewing experience–such as the effects of adding/removing *sound* rather than the L2 subtitles–have been seldom tested directly.

As a transient mode of verbal delivery, auditory input is generally more difficult to process than written input, especially for L2 learners [48], because it presents additional challenges such as accents, intonation, mode of delivery (e.g., mumbling or stuttering), multiple characters speaking at once, as well as volume changes (e.g., shouting vs. whispering), all of which can make it harder for L2 viewers to extract information from the auditory stream compared to L1 viewers [49]. For this reason, L2 subtitles are heralded as listening aids: by mapping aural word forms to their written counterparts [50, 51], they help L2 learners visualise what they hear, clarify ambiguous aural input, and decode the L2 speech stream into meaningful chunks [47, 52, 53], thus assisting L2 word identification as well as enhancing speech learning and language understanding [54]. However, it is also possible that, as long as L2 subtitles are present, L2 viewers will heavily rely on these and make less use of the audio, in effect using the subtitles as crutches [55, 56]. If this is the case, one would expect similar viewing patterns regardless of whether sound is present or not. If, on the other hand, L2 viewers actively use aural L2 input to support lexical processing and understanding when this channel is available, then one might expect a decreased reliance on the L2 subtitles–evidenced quantitatively in their reading patterns, for example through shorter subtitle processing times–when sound is present.

In the handful of eye-tracking studies that explicitly include a condition without sound [17, 22, 36, 38, 41], the majority did not examine L2 subtitle reading. Ross and Kowler [41] and d'Ydewalle et al. [36] did not consider L2 learners at all (only L1 speakers exposed to video with L1 audio and L1 subtitles). Łuczak [22] only considered subtitle reading in L1-Polish, and d'Ydewalle et al. [38] also only examined L1 subtitle reading (L2-German audio or no audio with L1-Dutch subtitles). However, reading in the L1 could have overriding effects [54] and because L2 processing is usually more difficult, less automatic, and slower than L1 processing [57], differences could be more visible when reading in the L2. To date, only Liao et al. [17] examined L2-subtitle reading with and without sound in some of their study conditions. In terms of eye movements in the subtitle area when watching video with L2-English subtitles, their non-native L1-Chinese viewers displayed significantly more fixations, longer saccades and less skipping without sound compared L1-Chinese audio. In the case of L2-English audio, although the same pattern was observed as in the case of the L1-Chinese audio, the difference between L2-English audio and no audio was only significant in the case of fixation count, making it difficult to draw definitive conclusions.

In the present study, we also examine L1-English processing in speakers of two distinct language varieties, Australian and British English, who are exposed to American English audio. Research has shown that, when different varieties of the same language are involved, accent can impede processing of information even for L1 speakers. For example, word processing has been shown to be impeded in the presence of regional or foreign accents as evidenced by delayed word identification [58]. This finding was robust for participants from the South-West of England when listening to either a French or Irish accent compared to a familiar Plymouth accent, and did not change with habituation. Similarly, L1 speakers of Canadian English were shown by Arnhold et al. [59] to be unable to make effective use of prosodic cues to disambiguate between two possible referents in British English instructions. In other words, word identification was again shown to be impeded by a regional accent (British English for Canadian English participants in this case). What this means for the current study is that speakers of Australian English and British English might also experience delayed auditory word identification resulting in more reliance on subtitles in the presence of American English audio, similar to the word identification difficulties experienced by the L2 viewers. Therefore, differences may be observed between the two L1 groups as well as between these two groups and the L2 group.

In summary, more research is needed to establish how the absence of sound affects L1 and L2 processing when watching subtitled videos, and whether L2 and L1 viewers use auditory input to different extents to support their understanding of the storyline. To address this gap, we use a variety of eye-tracking measures (see 'Eye movements' section) to explicitly compare the viewing patterns of L1 and L2 viewers, to assess whether and how the removal of sound affects L2-subtitle reading in comparison to L1-subtitle reading, as well as to determine whether there are differences between L1-subtitle reading by speakers of different language varieties (Australian and British English).

## The current study

Given evolving viewing habits, notably among younger generations who increasingly prefer streaming over traditional broadcast television [60], along with the proliferation of online services and social media platforms, where sound is often muted [11], our study seeks to explore how the absence of sound affects the overall viewing experience.

Additionally, we aim to investigate how different types of viewers–namely L1 speakers of the original language of the film and L2 viewers–engage with this novel type of viewing activity. With this goal in mind, the study examined three viewer groups: L1-English speakers from the United Kingdom and Australia, and L1-Polish speakers with advanced knowledge of English as a foreign language.

We asked two overarching research questions:

RQ1: *How does the absence of sound affect cognitive load, comprehension, recall, immersion, enjoyment and viewing patterns?*

RQ2: *Does the presence or absence of sound affect speakers of L1-British English, L1-Australian English and L2-English (L1-Polish) viewers differently?*

## Method

Adopting a mixed-methods approach [15], we collect quantitative data from self-reported immersion, enjoyment and cognitive load questionnaires, comprehension and recall post-tests, as well as eye-tracking data. Additionally, qualitative insights are gathered through semi-

structured interviews. Following Creswell & Creswell [15], we use the convergent mixed methods design, combining the results from the quantitative and qualitative data in order to see if the findings from both approaches confirm or disconfirm each other. Semi-structured interviews are also conducted for the purpose of completeness; i.e., to "bring together a more comprehensive account of the area of enquiry" [61].

Ethics clearance was secured from the ethics committees of each institution where data collection took place (Rector's Committee for the Ethics of Research Involving Human Participants at the University of Warsaw, Faculty of Arts and Humanities Research Ethics Subcommittee at the University of East Anglia, and Human Sciences Subcommittee at Macquarie University). Upon arrival at the laboratory, participants were provided with an information sheet about the study and had the opportunity to ask questions. After reviewing the information, participants signed a written consent form in order to participate in the study.

## Participants

A total of 168 participants took part in the study. Six people were excluded as they were not L1-English speakers of the two varieties under investigation–British and Australian English (N = 3) or not L1 English speakers at all (N = 3), and one person did not complete the full experiment. For comprehension, recall, self-reported cognitive load, immersion and enjoyment analyses, we collected usable data from 161 participants, of whom 43 were British ('UK'), 53 Australian ('AUS'), and 65 Polish ('PL'). Their mean age was 25.54 (SD = 8.80), ranging from 18 to 53. In the British cohort, the mean age was 26.37 (SD = 9.17), ranging from 18 to 45. In the Australian cohort, the mean age was 19.57 (SD = 1.06), ranging from 18 to 22. In the Polish cohort, the mean age was 29.86 (SD = 9.42), ranging from 19 to 53. Altogether we tested 115 females, 41 males, and five people who self-identified as non-binary. In the British cohort, there were 27 females, 12 males, and four non-binary participants. In the Australian cohort, there were 45 females and 8 males. In the Polish cohort, there were 43 females, 21 males and one non-binary participant. For the eye-tracking part, a further five participants were excluded due to poor-quality eye data, resulting in analysable data from 156 participants, of whom 63 were Polish, 42 were British and 51 were Australian.

When queried about how often they watch subtitled films and TV shows with the sound off on a scale ranging from 1 to 7 (where 1 = "never" and 7 = "always"), the overall mean rating was 1.57 (SD = 1.04). Specifically, for Polish participants' the average rating was 1.68 (SD = 0.97), for British participants 1.60 (SD = 1.15), and for Australian participants 1.40 (SD = 1.03).

Regarding the frequency of watching videos on social media without sound, the overall mean was 3.23 (SD = 1.54) on a 7-point scale (where 1 = "never" and 7 = "always"). Polish participants had an average rating of 3.34 (SD = 1.46), British participants 3.51 (SD = 1.76), and Australian participants 2.85 (SD = 1.40).

The descriptive statistics showed that Australians are the group that watches videos without sound the least frequently. However, the results of one-way ANOVAs for both questions indicated that the differences among the participants were not statistically significant in either of the two questions (p = .362 for films and TV shows, and p = .083 for social media).

When queried about watching English-language videos with English subtitles, on a scale ranging from 1 to 7 (where 1 = "least favourite way" and 7 = "most favourite way"), the overall mean was 5.74 (SD = 1.72). British participants had an average of 5.43 (SD = 1.91), Australian participants 5.55 (SD = 1.80), and Polish participants 6.11 (SD = 1.45), indicating high overall familiarity with the task. A one-way ANOVA showed that there was no statistically significant difference between the groups (p = .077).

We also measured the English proficiency of Polish participants with two tests: LexTALE [62] and Cambridge General English proficiency test freely available on the University of Cambridge English Language Assessment website. The mean LexTALE score was 79.22 (SD = 10.84), ranging from 53.75 to 100 (maximum). The mean score for the Cambridge test was 22.32 (SD = 3.08, ranging from 11 to 25 (maximum). The tests show that Polish participants had an average proficiency of C1 or C2 according to the Common European Framework of Reference for Languages (CEFR) and can therefore be regarded as proficient speakers of L2-English.

## Materials

Participants watched two videos with English-language subtitles, each lasting about 6 minutes, from the Netflix show *The Chair* [63]. American English was the language variety of the soundtrack and British English was the language variety of the subtitles, as this spelling is common to both L1 cohorts. One video was presented with sound, the other without. The two conditions were counterbalanced.

The subtitles used in the study were intralingual (i.e. English to English) and verbatim; that is, they included all the words from the dialogues. They did not contain any sound or speaker identifiers (unlike subtitles for the deaf and hard of hearing). Altogether, viewers were presented with 364 subtitles. The average speed at which subtitles were displayed was 16.85 characters per second (SD = 5.23). The mean number of characters per subtitle was 35.18 (SD = 17.14). See the replication pack for details.

## Apparatus

Eye movements were recorded with an EyeLink Portable Duo in the UK and an EyeLink 1000 Plus Eye Tracker (SR-Research, Canada) in Australia and Poland. In the UK, a target sticker was used for remote tracking with the Portable Duo mounted on an ASUS GX701L laptop, with participants sitting approximately 60 cm from the display screen (1920 x 1080 resolution). In Poland and Australia, a chinrest was used to minimise head movement, with participants sitting at approximately 80 cm from an ASUS VG248QE (16:9) HD monitor screen (1920 x 1080 resolution). The different optimised participant-to-screen distances ensured comparability, such that in all experimental setups, each subtitle letter subtended ~0.32˚ of the visual angle. Per the manufacturer's recommendations, a 13-point calibration was performed with the Portable Duo, and a 9-point calibration with the 1000 Plus. Data were recorded binocularly at a 1000 Hz sampling rate in all cases, and data were analysed for the right eye in most participants (N = 152, 97.4%). In a minority of cases (N = 4, 2.6%), as tracking accuracy was compromised only in the right eye, the left eye was used instead.

## Procedure

Data collection started in Poland on 29 November 2022 and finished in Australia on 26 May 2023.

Participants were tested individually in a research laboratory. They were pre-screened to be adult speakers of L1-Polish or L1-English. All participants signed an informed consent form, and Polish participants also took the English proficiency tests. All participants completed a demographics questionnaire. Then, they underwent calibration for the eye-tracking test. After watching each clip, they answered a set of comprehension, recall, cognitive load, immersion and enjoyment questions. Each question was displayed separately on the screen and participants had to click on the correct answer. At the end, participants took part in a 10–15-minute

semi-structured interview (see replication pack), which was audio recorded and transcribed for analysis.

## Design

We used a 2 x 3 mixed design with the presence or absence of sound (henceforth 'Sound') as the within-subject independent variable and language variety (henceforth 'Group') as a between-subject factor. In this design, 'Group' therefore represents the respective varieties of Australian L1-English, British L1-English, and L1-Polish.

We tested participants' comprehension, recall, cognitive load, immersion, and enjoyment. Comprehension was measured using ten true/false questions for each clip. Recall was measured with ten multiple-choice questions with four answers where only one answer was correct, and measured verbatim memory for numbers and exact wording. For comprehension and recall, we calculated the score as the percentage of correct answers.

Cognitive load was tested using three dimensions derived from the NASA-TLX instrument [64]: difficulty (equivalent to Mental Demand in the NASA-TLX instrument), effort (corresponding to Effort in the NASA-TLX instrument), and frustration (equivalent to Frustration Level in the NASA-TLX instrument). Immersion was measured using the captivation dimension [31]. Each dimension of cognitive load, immersion and enjoyment were measured on a 1–7 scale, where "1" represented the lowest value (smallest effort) and "7" represented the highest value (highest effort). We calculated the means for each condition per participant.

## Eye movements

Previous research has shown a direct link between cognitive processing and fixation durations, as longer fixations are believed to reflect increased attentional focus [65, 66] and more effortful word processing in reading [67]. In this study, we examine two fixation duration measures: *mean fixation duration* (MFD)–defined as the average duration of all fixations on the subtitle area–and *total reading time* (TRT)–defined as the sum of all fixation durations on the subtitle area. In reading, MFD is traditionally considered a measure of the depth of processing [42] and cognitive engagement with the text, with higher MFD values corresponding to more effortful processing. We examine this variable to enable a direct comparison with Liao et al. [17] and Łuczak [22], as per discussion above. TRT is believed to reflect later stages of lexical integration, as it includes regressive fixations and revisits to previously fixated words across all reading passes, possibly reflecting discourse-level integration [66]. By looking at this variable, we attempt to quantify the amount of overall attention to a subtitle.

We also consider *full skipping*, i.e., whether a subtitle was fixated at least once or not whilst on screen, and *crossover count* (vertical saccadic movements between the image and subtitle area) to investigate whether and how the presence of sound affects visual attention distribution between these areas. Together, these measures help establish whether, and if so, how, the need for subtitles is modulated by sound, and provide some insight into the global reading strategies adopted by the participants.

Regarding RQ1, we expect that the Sound-on condition might yield higher subtitle skipping rates and more crossovers, as well as lower TRT and MFD, as reading is less compelling when viewers can also hear what is written in the subtitles, since the processing of two different sources of information (auditory vs. visual) can occur in parallel. With regard to RQ2, we expect parallel processing to be less efficient in L2 populations (here: Polish participants), which should be reflected in longer fixation times (TRT) as well as less subtitle skipping and fewer crossovers than native viewers. However, as previous research has shown no significant differences in MFD in L1 and L2 subtitle reading [17, 22], we expect that this variable may or

may not be a significant predictor in the analyses. Moreover, due to potential differences in accent familiarity between Australian and British participants with the American English of the audio, we acknowledge that differences may emerge between these two groups (cf. [68]), although the direction of these differences is more difficult to predict as it is not the main focus of this study. We return to this point in the Discussion section.

## Data analyses

For comprehension and recall as well as self-reported data on cognitive load, enjoyment and immersion, we conducted analyses of variance (ANOVA) with Sound as the within-subject variable (with two levels: on and off) and Group as the between-subject factor (with three levels: UK, AUS, and PL). ANOVA analyses were conducted using SPSS version 29.

In the eye-tracking analyses, linear mixed-effect models (LMMs) were run for TRT, MFD and crossover counts. Generalised linear mixed-effect models (GLMMs) with a binomial family of distribution were run for subtitle skipping, as this is a binary outcome. The two key predictors of interest were Sound (on/off) and Group (AUS, UK, PL). All models included both fixed factors and their interactions. Fixed effects were known *a priori* and therefore entered first in the models. Then the maximal random effect structure (all possible intercepts and slopes) was modelled first, and progressively reduced until models started to converge. Different combinations of both correlated and uncorrelated slopes were fitted, and were kept only if they improved model fit [69, 70]. All final models included at least simple intercepts for both participants and items (i.e., subtitles). Model comparisons were carried out by means of ANOVAs and AIC/BIC comparisons, to reach a maximal yet parsimonious model [71].

In line with the literature, all individual fixations shorter than 60 ms and longer than 800 ms were excluded [72]. In the MFD and TRT analyses, all cases where MFD/TRT = 0 were also excluded, i.e., only cases where a subtitle was not skipped were included, as skipping was analysed separately.

MFD was log-transformed to improve the shape of the data. The eye-tracking analyses (see the replication pack) include a total of 156 participants and 364 subtitles. The number of data points included in the analysis varied depending on the variable considered: the MFD and TRT analyses comprised 36,591 observations, the skipping analysis 38,086 and the crossover analysis, 33,439. All models were run in R (version 4.3.1), using the default treatment coding, which compares each level of a categorical variable to a fixed reference, the intercept being the cell mean of the reference group [73]. In all analyses, the reference category for the Sound variable was the Sound-on condition, whereas for the Group variable it was the UK cohort, such that both AUS and PL are compared to the UK baseline. Bonferroni corrections for multiple comparisons were applied to avoid inflating the chance of false positive results [74], such that only p values below 0.0125 (0.05/4) were considered significant.

## Results

### Comprehension and recall

As shown in Tables 1 and 2, our 2 x 3 mixed ANOVA analyses revealed a statistically significant main effect of Sound on comprehension, $F(1,158) = 7.050$, $p = .009$, but not on recall, $F(1,158) = 3.858$, $p = .051$. Overall, participants had higher comprehension when they watched videos with sound ($M_{ON} = 81.30$) than without ($M_{OFF} = 77.70$). Similarly, recall was higher when the sound was on, $M_{ON} = 80.87$ vs. $M_{OFF} = 78.14$. However, the effect sizes for these results, as shown by partial eta squared ($\eta_p^2$) in Table 2, were small.

**Table 1. Estimated marginal means for comprehension and recall by Sound and Group.**

| | Sound | | | | | |
|---|---|---|---|---|---|---|
| | ON | | | OFF | | |
| | *M* | *SE* | *95% CI* | *M* | *SE* | *95% CI* |
| *Comprehension* (% correct answers) | | | | | | |
| UK | 82.09 | 1.96 | 78.21–85.97 | 82.55 | 2.21 | 78.17–86.93 |
| AUS | 80.94 | 1.76 | 77.45–84.43 | 75.28 | 1.99 | 71.33–79.22 |
| PL | 81.07 | 1.59 | 77.92–84.23 | 76.46 | 1.80 | 72.89–80.02 |
| *Recall* (% correct answers) | | | | | | |
| UK | 84.41 | 2.28 | 79.91–88.92 | 81.86 | 2.18 | 77.54–86.17 |
| AUS | 79.05 | 2.05 | 75.00–83.11 | 76.22 | 1.96 | 72.34–79.88 |
| PL | 80.00 | 1.85 | 76.33–83.66 | 77.23 | 1.77 | 73.72–80.11 |

There was no main effect of Group on either comprehension, F(2,158) = 1.856, p = .160, or on recall, F(2,158) = 2.884, p = .059. There were no interactions.

As shown in Fig 1, the highest comprehension overall was achieved by British participants ($M_{UK}$ = 82.32), followed by Polish ($M_{PL}$ = 78.76) and Australian participants ($M_{AUS}$ = 78.11). Comprehension was higher with than without sound for Polish and Australian participants, but slightly lower for British participants (see Table 1). However, the differences in comprehension between the participants were small and did not reach statistical significance (see Table 2).

The group with the highest overall recall scores was the British ($M_{UK}$ = 83.14), followed by Polish ($M_{PL}$ = 78.61) and Australian participants ($M_{AUS}$ = 77.64). When sound was taken away, recall scores dropped for all participant groups (see Fig 1). Similarly to comprehension, the differences between participants from different language varieties did not reach statistical significance.

We were also interested in whether Polish participants' recall and comprehension scores were related to their English proficiency. Indeed, both Cambridge proficiency and LexTALE scores correlated positively with recall and comprehension (see Table 3). This means that participants whose English proficiency was higher had higher comprehension and better recall. English proficiency did not correlate with any other dependent variable we measured, such as cognitive load, immersion, or enjoyment.

## Cognitive load

A 2 x 3 mixed ANOVA revealed statistically significant main effects of sound on all three indicators of cognitive load: difficulty (F(1,158) = 95.533, p < .001), effort (F(1,158) = 146.707, p < .001), and frustration (F(1,158) = 120.442, p < .001), see Tables 4 and 5. The effect sizes ($\eta_p^2$)

**Table 2. Two-way mixed ANOVA results for comprehension and recall.**

| | **df** $_{sound}$ | **df** $_{Group}$ | *F* | *p* | $\eta_p^2$ |
|---|---|---|---|---|---|
| Comprehension | | | | | |
| Between-subjects | | 2,158 | 1.856 | .160 | .023 |
| Within-subjects | 1,158 | | 7.050 | .009 | .043 |
| Recall | | | | | |
| Between-subjects | | 2,158 | 2.884 | .059 | .035 |
| Within-subjects | 1,158 | | 3.858 | .051 | .024 |

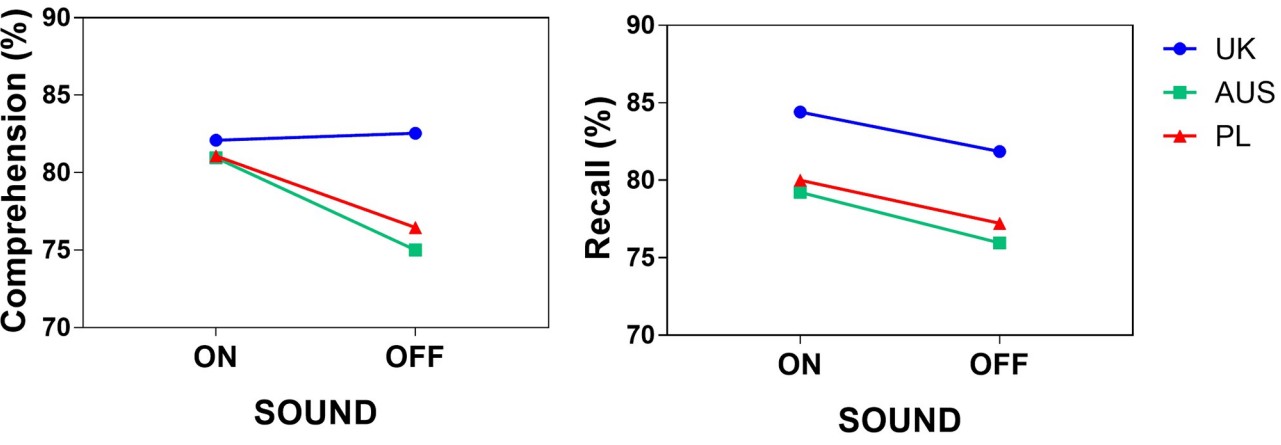

**Fig 1. Comprehension and recall by Sound and Group.**

for these effects were large, as shown in Table 5. In general, participants reported experiencing higher cognitive load in the absence of sound, as evidenced by higher mean scores for difficulty ($M_{ON}$ = 2.34 vs. $M_{OFF}$ = 3.70), effort ($M_{ON}$ = 3.07 vs. $M_{OFF}$ = 4.84), and frustration ($M_{ON}$ = 2.01 vs. $M_{OFF}$ = 3.63). These findings indicate that participants did not find watching the videos particularly difficult or frustrating, as indicated by mean scores below 4 on a 1–7 scale. However, they did perceive watching the video without sound as notably more effortful, with the mean score approaching 5 on a 1–7 scale.

In terms of between-group differences, we found a main effect of Group only in effort, (F (2,158) = 3.856, p = .023). As shown in Fig 2, the highest effort was reported by Australian participants ($M_{AUS}$ = 4.15) and the lowest by British participants ($M_{UK}$ = 3.52). Post-hoc Bonferroni tests showed that there was a statistically significant difference in effort between British and Australian participants, p = 0.023. Polish participants ($M_{PL}$ = 4.06) did not significantly

**Table 3. Correlations between English proficiency, comprehension and recall.**

| | Pearson Correlation | | | | |
|---|---|---|---|---|---|
| | 1 | 2 | 3 | 4 | 5 |
| 1. Cambridge test | - | | | | |
| 2. LexTALE | .549** | - | | | |
| | <.001 | | | | |
| 3. Comprehension Sound ON | .457** | .268* | - | | |
| | <.001 | .031 | | | |
| 4. Comprehension Sound OFF | .339** | .321** | .373** | - | |
| | .006 | .009 | <.001 | | |
| 5. Recall Sound ON | .433** | .392** | .378** | .359** | - |
| | <.001 | .001 | <.001 | <.001 | |
| 6. Recall Sound OFF | .274* | .349** | .250** | .338** | .317** |
| | .027 | .004 | <.001 | <.001 | <.001 |

Note:

**Correlation is significant at the 0.01 level (2-tailed),

*Correlation is significant at the 0.05 level (2-tailed)

**Table 4. Estimated marginal means for cognitive load indicators.**

| | Sound | | | | | |
|---|---|---|---|---|---|---|
| | ON | | | OFF | | |
| | *M* | *SE* | *95% CI* | *M* | *SE* | *95% CI* |
| *Difficulty*** | | | | | | |
| UK | 2.07 | .19 | 1.68–2.45 | 3.41 | .23 | 2.94–3.88 |
| AUS | 2.49 | .17 | 2.14–2.84 | 3.81 | .21 | 3.38–4.23 |
| PL | 2.40 | .16 | 2.08–2.71 | 3.78 | .19 | 3.40–4.16 |
| *Effort*** | | | | | | |
| UK | 2.65 | .21 | 2.23–3.06 | 4.39 | .24 | 3.91–4.87 |
| AUS | 3.24 | .19 | 2.87–3.62 | 5.05 | .22 | 4.62–5.49 |
| PL | 3.20 | .17 | 2.86–3.53 | 4.93 | .19 | 4.54–5.33 |
| *Frustration*** | | | | | | |
| UK | 2.09 | .18 | 1.73–2.45 | 3.23 | .25 | 2.72–3.74 |
| AUS | 2.01 | .16 | 1.60–2.34 | 3.88 | .23 | 3.42–4.34 |
| PL | 1.93 | .14 | 1.64–2.22 | 3.67 | .20 | 3.26–4.09 |

Note:

** measured on 1–7 scale where 1 was lowest and 7 was highest

differ from British or from Australian participants. There were no main effects of Group on neither difficulty nor frustration.

To check the internal validity of the three indicators of cognitive load (Difficulty, Effort, Frustration), we conducted the Cronbach's Alpha test. Cronbach Alpha for the sound-on condition was $\alpha = .73$, whereas for the sound-off condition $\alpha = .81$. Overall, the internal validity of our cognitive load measurement instrument was good.

## Immersion and enjoyment

A 2 x 3 mixed ANOVA showed a statistically significant main effect of Sound on immersion, ($F(1,158) = 36.854$, $p < .001$), with a large effect size (see Tables 6 and 7). Overall, participants reported higher immersion when the sound was on ($M_{ON} = 5.43$) compared to when it was off ($M_{OFF} = 4.73$).

**Table 5. Two-way mixed ANOVA results for cognitive load indicators.**

| | df <sub>sound</sub> | df <sub>Group</sub> | *F* | *p* | $\eta_p^2$ |
|---|---|---|---|---|---|
| Difficulty | | | | | |
| Between-subjects | | 2,158 | 1.721 | .182 | .021 |
| Within-subjects | 1,158 | | 95.533 | <.001 | .377* |
| Effort | | | | | |
| Between-subjects | | 2,158 | 3.856 | .023 | .047 |
| Within-subjects | 1,158 | | 146.707 | <.001 | .481* |
| Frustration | | | | | |
| Between-subjects | | 2,158 | .753 | .472 | .009 |
| Within-subjects | 1,158 | | 120.442 | <.001 | .433* |

Note:

*Large effect size

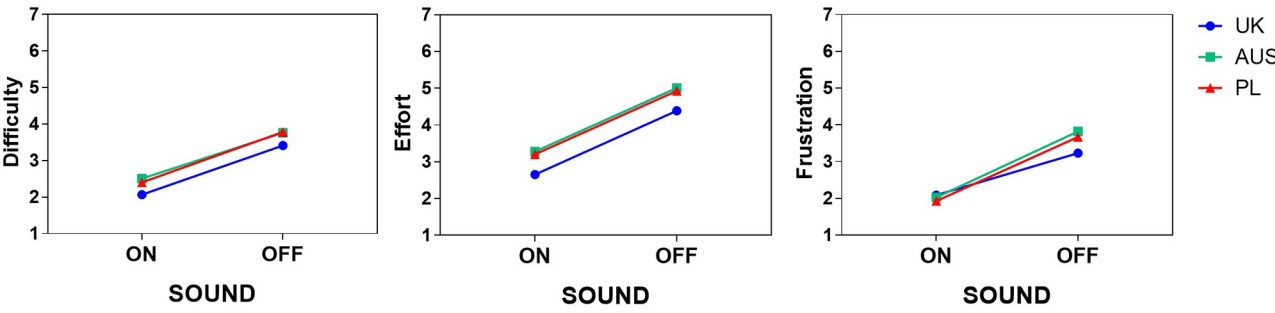

**Fig 2. Cognitive load results by Sound and Group.**

We also found a main effect of Group on immersion (F(2,158) = 4.809, p = .009). Overall, Australian participants experienced the lowest immersion ($M_{AUS}$ = 4.75) compared to British ($M_{UK}$ = 5.08) and Polish ($M_{PL}$ = 5.41) participants. Post-hoc Games-Howell tests showed that the difference was significant only between Polish participants and Australian participants, p = .005. There were no interactions.

Similarly to immersion, there was also a statistically significant main effect of Sound on enjoyment, (F(1,158) = 77.259, p < .001, see Tables 6 and 7). Overall, participants reported enjoying the videos more with sound ($M_{ON}$ = 5.52) than without sound ($M_{OFF}$ = 4.47).

There was no significant main effect of Group for enjoyment (F(2,158) = 2.095, p = .126). The lowest enjoyment was reported by Australian participants ($M_{AUS}$ = 4.72) compared to British ($M_{UK}$ = 5.11) and Polish ($M_{PL}$ = 5.13), but the difference was not statistically significant. Descriptive statistics showed that the sharpest drop in enjoyment when the sound was taken away was experienced by Polish participants and the smallest by British participants (see Fig 3).

We found a statistically significant interaction between Sound and Group (F(2,158) = 3.501, p = .033, eta = .042). We followed this up by conducting two one-way ANOVAs, one per each within-subject condition: one for Sound-on and one for Sound-off. We found a statistically significant difference in enjoyment between the participants when the sound was on, p = .021, but not when the sound was off, p = .192. When the sound was on, post-hoc Games-Howell tests showed that Polish participants experienced the highest enjoyment and differed

**Table 6. Estimated marginal means for immersion by Sound and Group.**

|  | Sound | | | | | |
|  | ON | | | OFF | | |
|  | *M* | *SE* | *95% CI* | *M* | *SE* | *95% CI* |
| *Immersion*** |  |  |  |  |  |  |
| UK | 5.27 | .18 | 4.92–5.63 | 4.88 | .22 | 4.43–5.33 |
| AUS | 5.22 | .16 | 4.90–5.50 | 4.28 | .20 | 3.87–4.69 |
| PL | 5.78 | .14 | 5.49–6.07 | 5.04 | .18 | 4.67–5.41 |
| *Enjoyment*** |  |  |  |  |  |  |
| UK | 5.41 | .18 | 5.06–5.77 | 4.81 | .24 | 4.33–5.29 |
| AUS | 5.22 | .16 | 4.90–5.55 | 4.22 | .21 | 3.79–4.65 |
| PL | 5.83 | .14 | 5.54–6.12 | 4.44 | .19 | 4.05–4.83 |

Note:

** measured on 1–7 scale where 1 was lowest and 7 was highest

Table 7. Two-way mixed ANOVA results for immersion and enjoyment.

| | df $_{sound}$ | df $_{Group}$ | F | p | $\eta_p^2$ |
|---|---|---|---|---|---|
| Immersion | | | | | |
| Between-subjects | | 2,158 | 4.809 | .009 | .057 |
| Within-subjects | 1,158 | | 36.854 | <.001 | .189* |
| Enjoyment | | | | | |
| Between-subjects | | 2,158 | 2.095 | .126 | .026 |
| Within-subjects | 1,158 | | 68.259 | <.001 | .302* |
| Interaction sound * Group | 2,158 | | 3.501 | .033 | .042 |

Note:

*Large effect size

from Australian participants, who experienced the lowest enjoyment, p = .013. When the sound was off, there were no significant differences between participants.

We also explored whether viewers' immersion and enjoyment were related to cognitive load (see Table 8). Firstly, we found that enjoyment was highly positively correlated with immersion: the higher the immersion, the higher the enjoyment reported by participants. Secondly, cognitive load indicators were correlated with one another; for example, the higher the difficulty reported by participants, the higher their frustration. Thirdly, immersion and enjoyment were negatively correlated with cognitive load: the higher the difficulty, effort, and frustration experienced by the participants, the lower their immersion and enjoyment.

## Eye tracking measurements

On average across all groups, in line with our hypotheses, mean fixation durations and total subtitle reading times were shorter with sound (MFD $M_{ON}$ = 202 ms; TRT $M_{ON}$ = 996 ms) than without (MFD $M_{OFF}$ = 209 ms; TRT $M_{OFF}$ = 1253 ms). Subtitle skipping occurred more frequently with sound ($M_{ON}$ = 0.055) than without ($M_{OFF}$ = 0.023), and a higher number of crossovers between the images and the subtitles were made with sound ($M_{ON}$ = 2.03) than without ($M_{OFF}$ = 1.96). Table 9 below reports descriptive statistics by Sound and Group, which are also presented graphically in Fig 4, showing that overall, the L2-English (PL) cohort

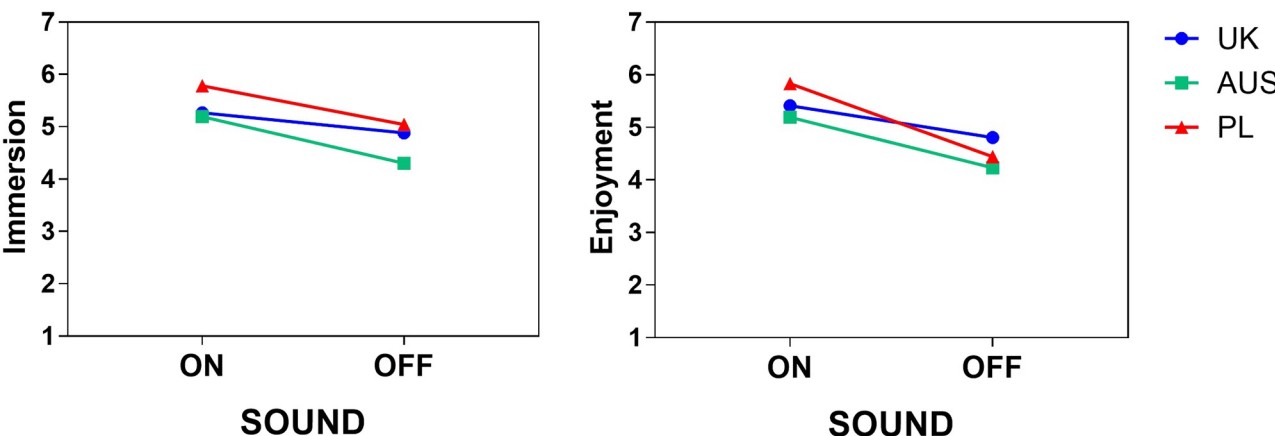

Fig 3. Immersion and enjoyment results by Sound and Group.

**Table 8. Correlations between enjoyment, immersion and cognitive load indicators.**

| | Pearson Correlation | | | | | | | | | |
|---|---|---|---|---|---|---|---|---|---|---|
| | 1 | 2 | 3 | 4 | 5 | 6 | 7 | 8 | 9 | 10 |
| 1. Enjoyment Sound ON | - | | | | | | | | | |
| 2. Enjoyment Sound OFF | .428** | | | | | | | | | |
| | <.001 | | | | | | | | | |
| 3. Immersion Sound ON | .858** | .384** | | | | | | | | |
| | <.001 | <.001 | | | | | | | | |
| 4. Immersion Sound OFF | .507** | .740** | .487** | - | | | | | | |
| | <.001 | <.001 | <.001 | | | | | | | |
| 5. Difficulty Sound ON | -.323** | -.075 | -.347** | -.177* | | | | | | |
| | <.001 | .344 | <.001 | .025 | | | | | | |
| 6. Difficulty Sound OFF | -.216** | -.429** | -.195* | -.415** | .302** | | | | | |
| | .006 | <.001 | .013 | <.001 | <.001 | | | | | |
| 7. Effort Sound ON | -.256** | -.066 | -.264** | -.145 | .494** | .310** | | | | |
| | .001 | .407 | <.001 | .068 | <.001 | <.001 | | | | |
| 8. Effort Sound OFF | -.135 | -.372** | -.205** | -.334** | .290** | .607** | .301** | | | |
| | .088 | <.001 | .009 | <.001 | <.001 | <.001 | <.001 | | | |
| 9. Frustration Sound ON | -.395** | -.087 | -.417** | -.166* | .495** | .053 | .497** | .103 | | |
| | <.001 | .273 | <.001 | .036 | <.001 | .503 | <.001 | .193 | | |
| 10. Frustration Sound OFF | -.171* | -.514** | -.222** | -.441** | .314** | .515** | .252** | .633** | .264** | |
| | .031 | <.001 | .005 | <.001 | <.001 | <.001 | .001 | <.001 | <.001 | |

Note:

**Correlation is significant at the 0.01 level (2-tailed),

*Correlation is significant at the 0.05 level (2-tailed)

appears to display different viewing patters from both L1-English cohorts. The gaze behaviour of the L1 cohorts looks roughly comparable in both sound conditions for TRT, skipping and crossover counts, as the values are generally close to each other, and the lines have similar slopes. However, for MFD, contrary to our predictions, it is not the PL cohort that displays the longest average fixation durations overall, but rather the AUS cohort. Table 10 presents model summaries for the four eye-tracking analyses, which are addressed in the next sections.

**Mean fixation duration.** The MFD analysis revealed a main effect of Sound ($b = 0.044$, $SE = 0.010$, $t = 4.196$, $p < 0.0001$), whereby watching without sound resulted in significantly higher MFD on the subtitles, thus mirroring the TRT results. However, unlike for TRT, there

**Table 9. Means and standard deviations of eye-tracking measures on the subtitle area by Sound and Group.**

| | Mean Fixation Durations (ms) | | | Total Reading Times (ms) | | | Skipping Rates | | | Crossover Counts | | |
|---|---|---|---|---|---|---|---|---|---|---|---|---|
| Sound | UK (SD) | AUS (SD) | PL (SD) | UK (SD) | AUS (SD) | PL (SD) | UK (SD) | AUS (SD) | PL (SD) | UK (SD) | AUS (SD) | PL (SD) |
| **ON** | 197 (60) | 207 (65) | 201 (55) | 922 (633) | 888 (623) | 1132 (724) | 0.074 (0.261) | 0.063 (0.243) | 0.036 (0.187) | 2.018 (0.830) | 2.122 (0.919) | 1.960 (0.875) |
| **OFF** | 205 (62) | 216 (70) | 207 (53) | 1165 (731) | 1167 (777) | 1379 (796) | 0.028 (0.165) | 0.029 (0.167) | 0.016 (0.124) | 1.976 (0.934) | 2.088 (1.032) | 1.840 (0.913) |

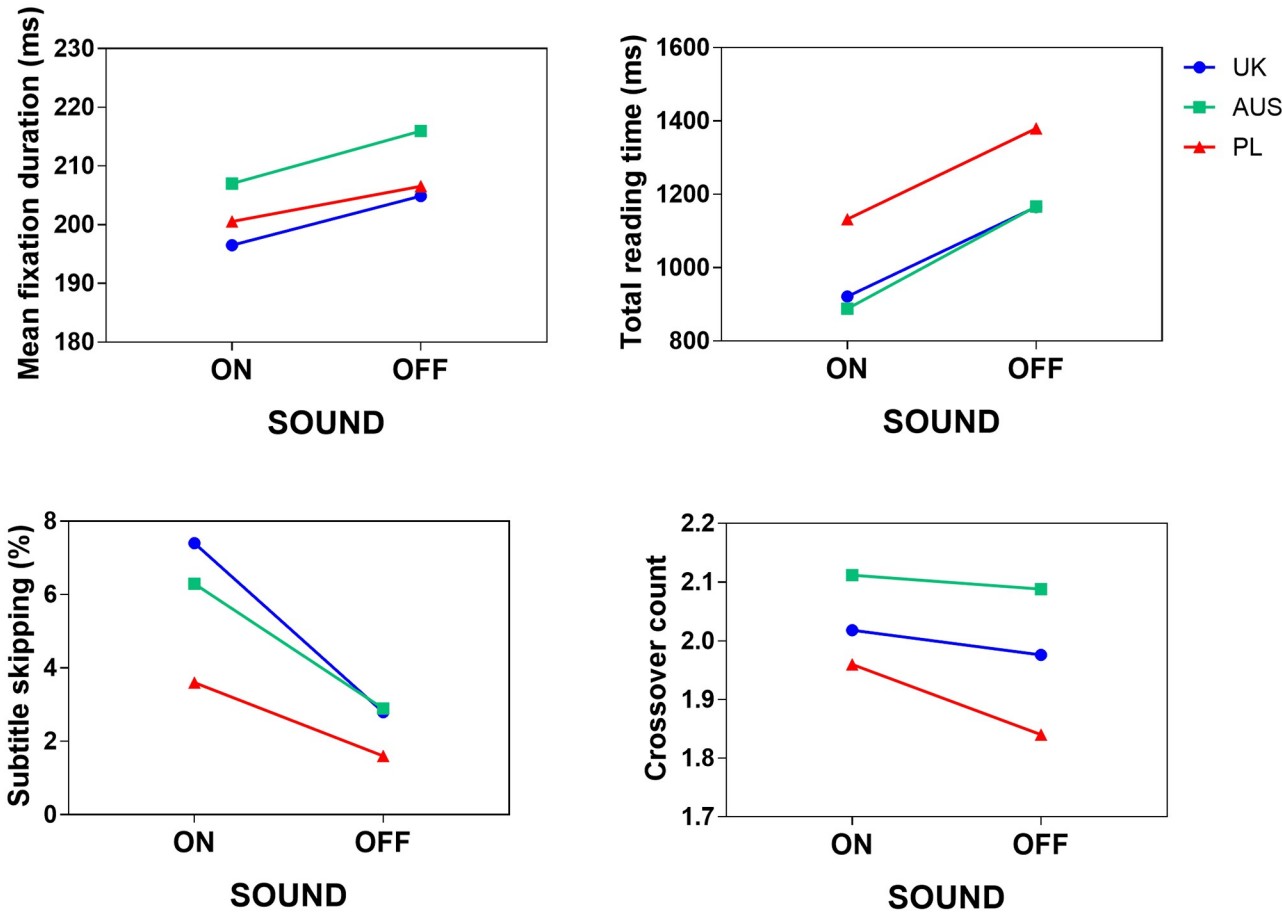

**Fig 4. Eye-tracking results by Sound and Group.**

was no main effect of Group, indicating that MFDs on the subtitles were comparable across cohorts. No significant interactions between Sound and Group were found (see Table 1).

**Total reading time.** A positive main effect of Sound was found ($b = 277.48$, $SE = 29.48$, $t = 9.413$, $p < 0.0001$), whereby watching without sound resulted in significantly longer reading times on the subtitle area. As expected, compared to the UK baseline, Group had a positive main effect for PL ($b = 233.68$, $SE = 42.73$, $t = 5.468$, $p < 0.0001$) but not for AUS ($b = -22.80$, $SE = 44.70$, $t = -0.510$, $p = 0.611$). These results indicate that L2 viewers spend significantly longer in the subtitle area compared to both L1 cohorts. No significant interaction between Sound and Group was found.

**Skipping.** The skipping analysis also revealed a main effect of Sound ($b = -1.192$, $SE = 0.107$, $z = -11.070$, $p < 0.0001$), with the negative sign of the coefficient indicating that viewers are significantly less likely to skip entire subtitles than without sound. A statistically significant effect of Group also emerged, whereby skipping rates for the PL cohort ($b = -0.907$, $SE = 0.247$, $z = -3.672$, $p = 0.0002$) are significantly lower than both native cohorts. Again, no significant interactions emerged.

**Crossover counts.** In the crossover analysis, unlike in all other analyses, no main effect of Sound was found. Regarding the number of crossovers by Group, the AUS cohort made significantly more visits to the images than the UK cohort ($b = 0.102$, $SE = 0.038$, $t = 2.667$,

**Table 10.  (G)LMMs results for the eye-tracking analyses on the subtitle region.**

| Measures | Contrasts | β | SE | t | p |
|---|---|---|---|---|---|
| Total Reading Time | (Intercept) | 883.20 | 43.31 | 20.390 | <.0001*** |
| | Sound (OFF) | 277.48 | 29.48 | 9.413 | <.0001*** |
| | Group (AUS) | -22.80 | 44.70 | -0.510 | 0.611 |
| | Group (PL) | 233.68 | 42.73 | 5.468 | <.0001*** |
| | Sound (OFF) × Group (AUS) | 18.02 | 38.38 | 0.469 | 0.639 |
| | Sound (OFF) × Group (PL) | -30.29 | 36.68 | -0.826 | 0.410 |
| Mean Fixation Duration | (Intercept) | 5.242710 | 0.020119 | 260.591 | <.0001*** |
| | Sound (OFF) | 0.044884 | 0.010697 | 4.196 | <.0001*** |
| | Group (AUS) | 0.042455 | 0.026614 | 1.595 | 0.113 |
| | Group (PL) | 0.025916 | 0.025442 | 1.019 | 0.310 |
| | Sound (OFF) × Group (AUS) | -0.003036 | 0.014198 | -0.214 | 0.831 |
| | Sound (OFF) × Group (PL) | -0.012810 | 0.013570 | -0.944 | 0.347 |
| Skipping Rates | (Intercept) | -3.54914 | 0.19925 | -17.813 | <.0001*** |
| | Sound (OFF) | -1.19266 | 0.10774 | -11.070 | <.0001*** |
| | Group (AUS) | -0.08445 | 0.25217 | -0.335 | 0.73771 |
| | Group (PL) | -0.90793 | 0.24724 | -3.672 | **0.00024***** |
| | Sound (OFF) × Group (AUS) | 0.19778 | 0.14396 | 1.374 | 0.16948 |
| | Sound (OFF) × Group (PL) | 0.19995 | 0.15647 | 1.278 | 0.20129 |
| Crossover Counts | (Intercept) | 1.993605 | 0.035201 | 56.635 | <.0001*** |
| | Sound (OFF) | -0.055121 | 0.033129 | -1.664 | 0.09822 |
| | Group (AUS) | 0.102937 | 0.038603 | 2.667 | **0.00854**** |
| | Group (PL) | -0.055338 | 0.036899 | -1.500 | 0.13587 |
| | Sound (OFF) × Group (AUS) | 0.008851 | 0.044669 | 0.198 | 0.84321 |
| | Sound (OFF) × Group (PL) | -0.081598 | 0.042733 | -1.910 | 0.05810 |

$p$ = 0.008). The PL cohort, on the other hand, did not differ significantly from the UK cohort ($b$ = -0.055, $SE$ = 0.036, $t$ = -1.500, $p$ = 0.135), although the negative sign of the coefficient confirms that L2 viewers make fewer crossovers than both L1 cohorts. No significant interaction emerged.

## Semi-structured interviews

**Watching without sound: Participant perceptions.** In the interviews after the experiment, a large majority of participants in all three countries reported that they found watching the subtitled videos without sound more difficult than with sound. However, a larger proportion of participants in Poland and Australia found this more difficult than those in the UK, where almost a third of participants said that they did not find it more difficult to watch the subtitled clips without sound. This is consistent with the cognitive load results, in which participants reported experiencing a higher cognitive load when the sound was absent, but the lowest effort was reported by British participants.

Many participants across all three countries commented that they found viewing the clips without sound more difficult because they had to concentrate more, and in particular because they had to focus more on the subtitles than on the action on screen, making it harder for them to follow the plot. Polish participant PL38 commented: "I think I paid attention mostly–if not only–to the subtitles. That was a difficulty because I had to focus on just one thing, and I couldn't watch what was really happening." Several participants remarked that, as a result, they

found it more challenging to keep up with the subtitles without sound, and that the lack of sound made the subtitles seem faster.

Moreover, a few participants stated that recall was harder with the clip without sound, with Australian participant AUS24 noting: "it was a lot harder to remember what the actual story-line was". Only a minority of participants said they did not find watching the subtitled videos without sound more difficult, including a larger proportion of British participants than in the other countries, as mentioned above. Some stated that they in fact found it easier to process and remember the clips, because they found that the lack of sound simplified the task, and they could read the subtitles more easily without the distraction of sound. A few participants commented on the impact of habituation and remarked that following became easier as the clip went on, as they became more accustomed to it.

Several participants noted that watching the videos without sound was more effortful because important information transmitted by the characters' tone of voice and inflection was missing. They remarked that the lack of sound made it more difficult for them to understand the emotions, intentions, and dynamics of interactions between characters, as well as the humour conveyed. Participant UK01 commented: "It can be difficult to distinguish tone and inflection, so you don't know if it's a shouted argument or a civil disagreement". Some participants also said that the lack of sound made it harder to identify which character was speaking (because the subtitles in this experiment were not meant for the deaf and hard of hearing, they did not include speaker labels or sound identifiers typically present in closed captions).

A few participants stated that they found watching the videos from the TV show *The Chair* without sound more challenging than viewing subtitled social media clips with no sound. One reason given was that, because social media videos are much shorter, much less concentration is needed, and another is the fact that "mindless scrolling" (AUS38) on social media platforms is a very different activity from sitting down to focus on a film or TV show. One participant also felt that watching subtitled videos without sound is easier on a smaller screen because of the shorter distance that the eyes must travel between the subtitles and the images: "I usually do this [watch without sound] on my smartphone, and it's easier to look at the entire picture and subtitles at the same time because it's smaller. And here, on the [computer] screen, you need to decide whether you look at the picture or at the subtitles. With tablets and smart-phones, it's easier, I think, because you just glance and you see it all, and with a big screen [. . .] it's really more complicated" (PL54).

In addition to finding the clips without sound more difficult to understand and process due to the absence of the intonation of characters' voices, the majority of participants also reported that watching without sound had a negative impact on their enjoyment, as they felt that some of the emotion and characterisation of the characters was missing. This is in line with the results of the immersion and enjoyment test, which found that participants' enjoyment was higher with sound. Moreover, participants missed the music and sound effects. Participant UK25 remarked, "I don't think [it's enjoyable] because you don't have the music and the sound effects. You only have what the person is saying. [. . .] It was less enjoyable because you couldn't hear the emotion and how people were talking".

Another common factor which not only made watching the clips harder for many partici-pants but also diminished their enjoyment was that they felt they had to make more of an effort to concentrate on the subtitles to the detriment of the time spent perusing the images. Partici-pant UK14 said: "It detracts from the experience, because you spend so much time trying to focus on what's going on instead of enjoying it" and participant PL42 remarked: "I was annoyed [. . .] I would focus on reading and I didn't see facial expressions or what was happen-ing in the background because I was focusing on the subtitles." Several participants also said

that it was less enjoyable because the experience was more comparable to reading a book than to watching a TV show.

While a couple of participants commented that they enjoyed watching the videos without sound more because they found it less distracting, and a minority said that the lack of sound did not affect their enjoyment, others expressed strong negative feelings about the experience. Several people said it was boring or frustrating, and some even felt it was "a waste of time" (AUS07), "a nightmare" (PL48), "torture" (AUS43), "soulless" (PL24), and participant UK29 felt that they "would have switched it off, not bothered watching at all". Immersion was also mentioned, with participant UK36 remarking that "It's a less immersive experience".

**Reasons for watching media content without sound.**   When questioned about when they would watch a video with the sound turned off, participants listed a wide variety of situations: public transport (when they do not have headphones), waiting rooms or other public places where having sound would be a nuisance to others, during classes if they are bored, at work, at the pub, at the gym (e.g., when running on a treadmill), at home to avoid waking sleeping children, or while multitasking (e.g., watching a video while listening to music). The variety of reasons for watching audiovisual content without sound enumerated by our participants aligns with previous commercial research conducted with American viewers [3, 8, 11]. Our participants also explained that they are able to concentrate on watching muted content only for a short period of time, with one person saying that they would only watch videos that are less than 20 seconds long without sound. Most interviewees also stated that this kind of viewing experience is rarely intentional, as they only do it if they have no choice. Some participants said that they usually choose videos that do not require having the sound on, especially if the subtitles or other forms of textual aid are available (e.g., clips with generic background music, cooking shows, sporting events, or news clips), with one person adding that they might sometimes watch TV shows with the sound off, but it must be something they are already familiar with. Additionally, many respondents reported on occasion watching videos with the volume very low so as not to disturb other people, especially late at night. Finally, most interviewees stressed that they would only watch muted videos on their smartphones or tablets, and not on larger screens, such as TV or laptops, because–as noted by participant PL54 –"it's easier to look at the entire picture and subtitles at the same time because it's smaller".

## Discussion

The aim of this study was to investigate how the presence or absence of sound influences viewers' comprehension, recall, cognitive load, immersion, enjoyment, and eye movements. Furthermore, we explored how the impact of sound affects viewers of different language varieties by considering three participant groups: L1-Polish speakers with high L2-English proficiency, along with L1-English speakers of British English and Australian English. Using a convergent mixed methods approach [15], we triangulated the quantitative data with the qualitative assessment of people's reasons for watching audiovisual content without sound collected via in-depth interviews. The qualitative findings also provided a more comprehensive account of participant perceptions.

### The impact of sound

A consistent pattern of results emerged overall, indicating that the absence of sound had a negative effect on all aspects of the viewing experience as assessed in the questionnaires: comprehension, recall, cognitive load, enjoyment, and immersion. Across all measures, when the sound was turned off, participants' level of comprehension, recall, enjoyment and immersion significantly diminished in comparison to when the sound was on. Furthermore, participants

reported experiencing higher cognitive load, particularly in terms of the effort required to watch, when the sound was absent.

It is important to acknowledge that while the removal of sound did indeed have an adverse effect on comprehension, the magnitude of these findings was relatively modest, as indicated by the small effect size reported in the tests. For example, when sound was present, comprehension was slightly higher overall at 81%, but the condition without sound yielded 78% accuracy, so participants' understanding was still relatively high. In a broader perspective, this difference does not appear particularly substantial. Therefore, although the overall viewing experience is more complete, immersive and enjoyable with sound, the absence of sound does not reach a level of detriment that would generally dissuade viewers from engaging in this type of experience. Moreover, recall was not significantly affected by Sound, which shows that participants were able to recall exact words and expressions equally well with and without sound.

Unlike in the case of comprehension and recall, where the mean differences between watching videos with and without sound were relatively minor and their effect sizes were small, the differences in cognitive load, immersion and enjoyment were more substantial, with large effect sizes. In general, participants experienced lower immersion and enjoyment as well as higher cognitive load without sound. We also found that these effects correlate: the higher the cognitive load reported by participants, the lower their immersion and enjoyment, which is in line with our expectations.

The removal of sound led to an increased cognitive load, although not all cognitive load indicators were affected equally. Notably, the one most significantly affected was effort, which was highest in the condition without sound across all participant groups. Interestingly, when we take into account the comprehension and recall findings, although participants exerted greater effort, they were able to attain similar levels of comprehension and recall. It appears, therefore, that participants were able to effectively handle the task demands created by taking away the sound when watching the videos used in this study. It remains possible, however, that the cognitive effort required to follow longer videos over an extended period may potentially lead to fatigue–an effect we could not assess in the current design, which used relatively short video clips.

In terms of the cognitive load theory [20], the findings of this study confirm the hypothesis proposed by Sweller and colleagues [75]. Our results support the multimodal and modality effects as well as the split attention postulated in the integrated model of text and picture comprehension [16]. The removal of the auditory channel resulted in increased cognitive load as all processing had to be performed within the visual channel. Additionally, our comprehension results also demonstrate a significant decrease in comprehension (albeit with small effect sizes, indicating that comprehension loss without sound is not that substantial, but it nevertheless occurs).

The absence of sound also had an adverse effect on viewers' immersion in the story world and their suspension of disbelief. In other words, the viewers' capacity to "enter into and become engaged with the circumstances and the inhabitants of fictional worlds" [24] was compromised when sound was removed. A factor that is known to positively influence immersion is perceived realism [76]–the more realistic the viewing experience, the greater the potential for immersion. As succinctly expressed by participant PL44, "Life has sound," so it is logical to conclude that the presence of sound leads to higher realism and viewing without sound therefore results in lower immersion. We note that our study focused on fictional narratives, which, on the one hand, typically induce higher levels of immersion than non-fictional genres, but, on the other hand, are commonly viewed with sound, unlike short-form social media videos. Furthermore, existing literature suggests a link between immersion and enjoyment: when viewers seek to maximise enjoyment, they "actively stop evaluating the authenticity of fictional

content" [24]. Our study provides quantitative evidence for this relationship, where immersion showed a positive correlation with enjoyment.

The results also showed that viewers derived significantly less enjoyment from the videos without sound. Prior research has shown that viewers' sentiments towards film characters and their overall enjoyment hinge on "a viewer's affective disposition towards characters" [77]. The ability to hear film characters speak enables viewers to connect with the characters' emotions, fostering an affective disposition towards them. This was confirmed by several participants, including AUS33, who said: "Obviously sound makes it more entertaining and you get to hear the emotions when they talk", and participant UK25, who similarly observed, "it was less enjoyable because you couldn't hear the emotion".

The role of sound in film extends to the accompanying mood music, influencing the emotions elicited during film viewing. Without sound, self-identification with characters and the formation of an emotional bond with them becomes more challenging for viewers, resulting in diminished enjoyment. Indeed, the interviews echoed the findings from the questionnaires, as participants described the videos without sound as "soulless" and "bland," thus confirming that the absence of sound contributes to a less engaging and emotionally resonant viewing experience. We additionally note that the types of audiovisual content participants typically watch without sound are short-form videos on social media, where there is less chance to form an emotional bond than in TV shows with a more complex narrative and story world. To sum up, the results consistently showed that the absence of sound had a negative impact on various aspects of the viewing experience, as perceived by the participants.

In line with our expectations, in terms of the eye-tracking analyses the absence of sound also had a measurable effect across nearly all eye movements examined. Taking away the sound resulted in significantly longer mean fixation durations and total reading times, as well as lower skipping rates on the subtitles, indicating that all participants focused their visual attention on this area when watching without sound. This is relatively understandable if we consider that, without sound, the subtitles are the only source of verbal information enabling viewers to follow the storyline–resulting in more thorough text reading in this condition. The quantitative eye-tracking results with respect to sound thus confirm the questionnaire analyses and the qualitative data emerging from the interviews. They are also aligned with other studies where the presence of sound was shown to affect subtitle reading by decreasing total reading time [17, 38] and resulted in more subtitle skipping [17, 41].

Interestingly, however, and contrary to our expectations, the crossover analysis did not reveal a statistically significant effect of sound: participants shifted attention between the subtitles and the images in comparable amounts in both sound conditions. The reasons for this finding could be manifold. First, some participants found speaker identification challenging without sound. This could have triggered compensatory crossovers to the images and back to the text to try and identify which characters the subtitle words were being uttered by, especially in a dialogue-heavy TV scenes featuring several participants conversing on screen. Interviews revealed that participants felt they were not processing facial expressions and other details in the images without sound, reinforcing the need for attention shifts. Moreover, participants also mentioned the shorter distance that the eyes must travel between the subtitles and the images on mobile devices (which is where they usually watch audiovisual content without sound) compared to the much larger monitor screen used in the experiment. Participants feeling that watching without sound became easier as the clip progressed may suggest a flexible application of compensation strategies, particularly in cases of unclear speaker identification from written verbal input alone. This adaptability could have contributed to masking the overall trend that is nevertheless visible in the data, i.e., that fewer crossovers are made without sound, reflecting an increased reliance on the subtitles.

Overall, removing the soundtrack affected the viewing experience in significant ways, increasing the reliance on subtitles. This was evidenced through increased inclination to read them (as indicated by less skipping) and a more thorough processing evidenced by longer fixation durations, both in total and on average. In consequence, the images were visited less often (as evidenced by lower crossover counts), but not significantly so, as viewers could and did redirect attention to the images if they felt that they needed to. By triangulating eye movements, comprehension, and interview data it is possible to conclude that, in the absence of sound, viewers adapt their gaze using metacognitive strategies to maintain comprehension. This involves not only reading the subtitles more thoroughly, but also referring to the images when character identification is challenging or when looking at character faces aids in deducing emotions, intentions, or the interactional dynamics crucial for understanding the unfolding action.

## Watching with and without sound in L1 and L2 contexts

When it comes to differences between the three groups, we expected that Polish participants, as L2-English speakers, would face greater challenges with the task, leading to higher cognitive load and reduced levels of comprehension, recall, enjoyment, and immersion in self-reported measures. However, contrary to our hypothesis, there were no statistically significant differences between language groups in terms of comprehension, recall, or enjoyment. While British participants did indeed report the lowest level of effort, it was the Australian participants, rather than Polish, whose self-reported effort was the highest. Regarding immersion, a significant difference emerged between L1 and L2 cohorts but only insofar as Australian viewers experienced significantly lower immersion than Polish viewers. The fact that Polish viewers did not report the highest effort or lowest immersion could be attributed to the fact that they were highly proficient in English and familiar with the nature of the task of watching L2-English-language content with L2-English subtitles. Indeed, previous research has indicated that being accustomed to subtitling could influence viewers' perception and experience of the task; for instance, in a study by Jensema [78], those less familiar with subtitling tended to feel less at ease with faster subtitle speeds. Moreover, although Australian participants were younger than both British and Polish participants, and younger generations are generally more used to watching subtitled videos without sound [8], this was not the case in the young Australian cohort examined in the present study, whose ratings regarding familiarity with watching without sound were the lowest (see 'Participants' section).

When it comes to the eye-tracking analyses, our hypotheses were largely confirmed. A main effect of language emerged in the total reading time and skipping analyses, as the Polish cohort spent considerably longer reading subtitles and tended to skip them less compared to the L1-English cohorts both with and without sound. This aligns with the findings presented by Liao et al. [17], who also investigated viewing behaviour in L1 and L2 both with and without sound. Our findings overall confirm their results and show that reading in a second language, despite proficiency in that language, remains slower than reading in one's native language. They also suggest that having the soundtrack available in one's native language (as was the case for British and Australian viewers) makes processing more effective compared to when it is available in a second language (Polish viewers).

In terms of mean fixation duration, which is an indicator of cognitive load and depth of processing [42], Australian participants had the highest MFD among all groups, although this difference did not reach statistical significance. The highest cognitive effort registered via eye movements therefore corresponds to the highest cognitive effort as self-reported by the Australian participants. The lack of statistical significance in this difference is not unexpected,

given the highly idiosyncratic individual differences in MFD scores among people [42]. It is also consistent with prior research findings, which did not find statistical differences in MFD [17, 22].

In the crossover analysis, we found an unexpected effect of language variety, as it was the Australian–and not Polish–participants who differed significantly from British participants, in that they made more visits to the images regardless of whether sound was present or absent. The only significant differences found between the two L1 cohorts are therefore in terms of crossovers and self-reported effort. One explanation for these differences between Australian and British participants could be familiarity effects with the American accent of the video, as research has shown perceivable differences between English varieties. For example, cross-entropy–a measure of the differences across language varieties–between American and Australian English is 25.8, between American and British English is 36.8, and between British and Australian 2.9 [68]. These values not only highlight the distinctive nature of these varieties, but also emphasise that both British and Australian varieties are relatively distant from the American one. In principle, therefore, if a viewer is unfamiliar with this relatively distant L1 language variety, they might process information more slowly or scrutinise images to confirm details, such as faces and lips. Given that different pronunciation and intonation patterns can affect speech perception even in L1 speakers [58, 59], it is possible that the L1-Australian cohort's potential unfamiliarity with the language variety in the dialogues could have increased the effort required to follow the dialogues. This, in turn, may have contributed to a higher frequency of shifts towards the images and a greater self-reported effort compared to L1-British participants. Finally, the idiosyncratic reading behaviour of the Australian cohort could also be partially due to individual differences in how they approached and how much they appreciated the whole viewing experience. The self-reports analysed above clearly show that they reported not only the highest effort, but also the lowest immersion and enjoyment, as well as the lowest comprehension and recall scores. Therefore, as reiterated in the interviews, Australian participants found the experience–especially without sound–more difficult than British participants overall. This could have affected how often they went back to the images and be reflected in the highest self-reported effort scores.

## Conclusion, limitations and future research

This mixed methods study investigated how sound influences viewers' comprehension, recall, cognitive load, immersion, enjoyment, and eye movements. Additionally, we examined how the presence or absence of sound impacts viewers from different language varieties in three participant groups: Polish speakers with high L2-English proficiency and L1-English speakers of British English and Australian English. Both quantitative and qualitative strands of the study revealed that when watching subtitled videos without sound, viewers experienced higher cognitive load, lower immersion and enjoyment, and somewhat reduced comprehension and recall. The absence of sound significantly affected the viewing experience by increasing the viewers' inclination to read and process subtitles more thoroughly. In terms of reading behaviour, L2 participants spent more time reading the subtitles and skipped fewer subtitles than the L1 groups.

We also need to acknowledge some limitations of the study. The three participant cohorts involved differed in terms of age as well as familiarity with subtitling: Australian participants tended to be younger and less experienced with subtitling compared to participants from the UK and Poland. This demographic disparity could potentially introduce a confounding factor. Factors like participants' multilingualism and reading skills were not systematically controlled

for, which is another limitation that needs to be acknowledged and limits the generalisability of the findings.

Despite these limitations, however, the present study shows that removing sound has the potential to exert top-down influences across all participants regardless of language variety, altering both subtitle reading behaviour and the overall film viewing experience. The addition of sound leads to a significant decrease in the time spent on subtitles and an increase in skipping, suggesting active engagement with the soundtrack. The results show that sound modulates the need for subtitles, which become less essential when sound is present.

This study focussed on reading English-language subtitles in an English-language TV show. More research is needed with other language combinations, and particularly looking at subtitle *translations*–i.e., cases when the written input is not a near word-by-word *transcription* of the audio input but involves content transfer from another language–as this may also affect reading patterns. Studies investigating the influence of accents on subtitle reading could also provide more information on whether familiarity with a given spoken language variety would impact eye movements when reading subtitles.

On a related note, we also acknowledge that in this study reading behaviour was only examined at the subtitle level. In the future, including more nuanced, word-level measures will be useful in examining reading strategies with and without sound, and assessing whether different classes of words are affected differently. Another fruitful avenue for research is investigating how the absence of sounds affects processing of the moving *images*, which constitute a key yet relatively understudied component of watching subtitled films. Finally, in future research, it would also be valuable to investigate the effects of the absence of sound on the viewing experience with clips of different lengths and genres. On the one hand, short-form video content on social media platforms deserves more attention, as this appears to be the context in which watching without sound most often occurs. On the other hand, examining longer video materials such as complete episodes would be useful to assess whether other effects such as fatigue come into play, as this might have more pronounced effects on comprehension, cognitive load, immersion, and enjoyment, whilst eye movement patterns could also reflect different or additional viewing strategies.

## Supporting information

**S1 Data.**
(ZIP)

## Author Contributions

**Conceptualization:** Agnieszka Szarkowska, Sonia Szkriba, David Orrego-Carmona, Jan-Louis Kruger.

**Data curation:** Agnieszka Szarkowska, Valentina Ragni, Sonia Szkriba, Sharon Black, David Orrego-Carmona.

**Formal analysis:** Agnieszka Szarkowska, Valentina Ragni, Sharon Black.

**Funding acquisition:** Agnieszka Szarkowska.

**Investigation:** Agnieszka Szarkowska, Valentina Ragni, Sonia Szkriba, Sharon Black, David Orrego-Carmona.

**Methodology:** Agnieszka Szarkowska, Valentina Ragni, David Orrego-Carmona, Jan-Louis Kruger.

**Project administration:** Agnieszka Szarkowska, Sharon Black, Jan-Louis Kruger.

**Resources:** Agnieszka Szarkowska, Sonia Szkriba.

**Software:** Agnieszka Szarkowska, Valentina Ragni.

**Supervision:** Agnieszka Szarkowska.

**Validation:** Valentina Ragni.

**Visualization:** Valentina Ragni.

**Writing – original draft:** Agnieszka Szarkowska, Sonia Szkriba, Sharon Black.

**Writing – review & editing:** Agnieszka Szarkowska, Sonia Szkriba, Sharon Black, David Orrego-Carmona, Jan-Louis Kruger.

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
