## [Decision Letter · Decision Letter 0]

15 Feb 2024

PONE-D-23-41429Watching subtitled videos with the sound off affects viewers’ comprehension, cognitive load, immersion, enjoyment, and gaze patterns: A mixed methods eye-tracking studyPLOS ONE

Dear Dr. Szarkowska,

Thank you for submitting your manuscript to PLOS ONE. After careful consideration, we feel that it has merit but does not fully meet PLOS ONE’s publication criteria as it currently stands. Therefore, we invite you to submit a revised version of the manuscript that addresses the points raised during the review process.

Your manuscript has now been peer reviewed. Reviewers asks for a different approach to analyse and interpret your data. Please could you follow their recommendations (reviewers number 2 and 3) and submit a revised version with a new re-analysis for our consideration?==============================

We look forward to receiving your revised manuscript.

Kind regards,

Antonio Peña-Fernández, PhD

Academic Editor

PLOS ONE

Journal Requirements:

2. Thank you for submitting the above manuscript to PLOS ONE. During our internal evaluation of the manuscript, we found significant text overlap between your submission and previous work in the [introduction, conclusion, etc.].

Please revise the manuscript to rephrase the duplicated text, cite your sources, and provide details as to how the current manuscript advances on previous work. Please note that further consideration is dependent on the submission of a manuscript that addresses these concerns about the overlap in text with published work.

[If the overlap is with the authors’ own works: Moreover, upon submission, authors must confirm that the manuscript, or any related manuscript, is not currently under consideration or accepted elsewhere. If related work has been submitted to PLOS ONE or elsewhere, authors must include a copy with the submitted article. Reviewers will be asked to comment on the overlap between related submissions (http://journals.plos.org/plosone/s/submission-guidelines#loc-related-manuscripts).]

We will carefully review your manuscript upon resubmission and further consideration of the manuscript is dependent on the text overlap being addressed in full. Please ensure that your revision is thorough as failure to address the concerns to our satisfaction may result in your submission not being considered further.

"This study was conducted within the WATCH-ME project funded by the National Science Centre OPUS 19 programme 2020/37/B/HS2/00304. The principal investigator is Agnieszka Szarkowska, whereas Valentina Ragni is employed as a post-doc and Sonia Szkriba as a PhD candidate researcher in the project."

Additional Editor Comments:

Dear authors,

Thank you for considering our journal for publishing your research. Your manuscript has now been peer reviewed and I agree with comments provided, particularly with those provided by reviewers number 2 and 3. I consider that your methods and analysis should be properly enhanced so we can consider it for publication. Could I ask authors to comprehensively review these aspects of your manuscript and provide a new re-analysis of your data following the reviewers recommendations, please?

Best wishes,

Antonio

Reviewers' comments:

Reviewer's Responses to Questions

**Comments to the Author**

1. Is the manuscript technically sound, and do the data support the conclusions?

Reviewer #1: Yes

Reviewer #2: Partly

Reviewer #3: Yes

2. Has the statistical analysis been performed appropriately and rigorously? 

Reviewer #1: I Don't Know

Reviewer #2: No

Reviewer #3: Yes

3. Have the authors made all data underlying the findings in their manuscript fully available?

Reviewer #1: Yes

Reviewer #2: Yes

Reviewer #3: Yes

4. Is the manuscript presented in an intelligible fashion and written in standard English?

Reviewer #1: Yes

Reviewer #2: Yes

Reviewer #3: Yes

5. Review Comments to the Author

Reviewer #1: - Overall, this is a good paper about Watching subtitled videos with the sound off affects viewers’ comprehension, cognitive

load, immersion, enjoyment, and gaze patterns.

- • The introduction and presentation of the problem do not properly fulfil their roles: they do not introduce or appropriately address the issues of the research.

- • The research problem is not clear, and it needs to be supported and explained why the researchers did the research.

- • How general are your results? These have to be of interest to the whole communityز

- • Interpretations of the results should be provided in light of previous studies and training theoriesز

- • No relevant theories were included in the discussion to support or provide anchorage to the research problem.

Many thanks to the study team for the remarkable research they conducted and the unique manner in which it was presented.

Reviewer #2: 1.The paper frequently employs long sentences, and breaking some of them would improve overall comprehension.

2.Elaborate on how the sample size was estimated (justification).

3.Sampling technique employed (how) and related sampling bias should be discussed.

4.The justification of why different apparatuses were used for eye movement measurements, as well as differing distances from the screen and potential biases, should be elaborated on.

5. Highlight on the study duration and period it was carried out.

6. The study topic under study could be influenced or differ by sex or gender. Any sex-related analysis in any of the analyses performed could have provided deeper insights. Authors should justify why such related analyses were not considered as well as why they did not consider controlling for “any variable” that potentially may have influenced the relations (for instance, how are we sure the differences observed are not more due to age, education background, familiarity with technology, or subtitles other than absence of sound?).

7.To strengthen the study findings, the authors should at least consider including potential confounding variables whose data was collected in some of the analysis. Recognising this as a limitation is not a good practice, especially when the data is available (i.e., trustworthy is limited).

8. Even though the paper did a detailed literature review, the introduction or background appear to be too long..

Reviewer #3: Accepted with Major revisions. Triangulation for data analysis is compulsory for Mixed methods studies. Details of questionnaire was not defined properly, was it a self-designed by the research team of the project or adopted?

Same observation holds for semi-structured interview.

NOTE: If both instruments were adopted then whether reliability and validity of both were tested in the given research context. I don’t find such information in any part of the part of the paper. If not then it must be tested and validated.

6. PLOS authors have the option to publish the peer review history of their article (what does this mean?). If published, this will include your full peer review and any attached files.

Reviewer #1: No

Reviewer #2: No

Reviewer #3: No

---

## [Author Response · Author response to Decision Letter 0]

1 Mar 2024

We have attached a PDF with a response to Reviewers and Editors. We are pasting them here too, but please refer to the PDF for better readability.

Response to Reviewers

Manuscript: Watching subtitled videos with the sound off affects viewers’ comprehension, cognitive load, immersion, enjoyment, and gaze patterns: A mixed methods eye-tracking study

First, we would like to start by expressing our gratitude to the Reviewers for their examination of our manuscript and for the valuable feedback they provided. We trust that their suggestions have contributed to enhancing our manuscript. 

All changes can be located in the attached word file with tracked changes. We also attached a clean version, with all changes accepted. Below, we address specific issues raised by the Editors first and then by each reviewer. 

All reviewers’ and editors’ comments are marked in blue. The changes we introduced in the manuscript are reported in green.

Editors

We have gone through the file naming conventions and confirm that they manuscript meets the requirements.

[Comment 2 disregarded, following the correspondence with Teresa Diviacchi, Publishing Editor, on 21 February 2024].

"This study was conducted within the WATCH-ME project funded by the National Science Centre OPUS 19 programme 2020/37/B/HS2/00304. The principal investigator is Agnieszka Szarkowska, whereas Valentina Ragni is employed as a post-doc and Sonia Szkriba as a PhD candidate researcher in the project."

The funder did not have any role in the study, so we confirm that this statement is true: "The funders had no role in study design, data collection and analysis, decision to publish, or preparation of the manuscript."

We have included the full ethics statement in the first version of the manuscript we submitted, including the full names of the ethics committees and the fact that the consent was written. Here is the relevant fragment:

Ethics clearance was secured from the ethics committees of each institution where data collection took place (Rector’s Committee for the Ethics of Research Involving Human Participants at the University of Warsaw, Faculty of Arts and Humanities Research Ethics Subcommittee at the University of East Anglia, and Human Sciences Subcommittee at Macquarie University). Upon arrival at the laboratory, participants were provided with an information sheet about the study and had the opportunity to ask questions. After reviewing the information, participants proceeded to sign a written consent form in order to participate in the study.

All the documentation related to ethics, including the clearances and their reference numbers, have been submitted together with the manuscript.

5. Thank you for considering our journal for publishing your research. Your manuscript has now been peer reviewed and I agree with comments provided, particularly with those provided by reviewers number 2 and 3. I consider that your methods and analysis should be properly enhanced so we can consider it for publication. Could I ask authors to comprehensively review these aspects of your manuscript and provide a new re-analysis of your data following the reviewers recommendations, please?

Thank you for considering our manuscript for publication in PLoS ONE. We appreciate all the comments and suggestions, and we address each of them below.

As requested, we have re-run our analyses to include gender, age, and years of education. We have also calculated Cronsbach alpha to validate the internal consistency of the cognitive load instrument we used. We report the details below.

Reviewer 1

- Overall, this is a good paper about Watching subtitled videos with the sound off affects viewers’ comprehension, cognitive load, immersion, enjoyment, and gaze patterns.

- The introduction and presentation of the problem do not properly fulfil their roles: they do not introduce or appropriately address the issues of the research. The research problem is not clear, and it needs to be supported and explained why the researchers did the research.

We have rewritten the Introduction to better present the research problem and added a section on Theoretical Framework (see below).

- How general are your results? These have to be of interest to the whole communityز

We believe our results are relevant to viewers across the globe, who increasingly engage in watching videos, especially on social media, with the sound turned off. This global trend, confirmed by a number of publications coming from different countries (some of which we list below), calls for research on how the new watching habits affect viewers. 

Kanter, J., Young people use subtitles to get the plot, in The Times. 2021. p. 20.

Pulse The Majority of Young People Are Using Subtitles When They Watch TV. YPulse, 2022.

Zajechowski, M., Survey: Why America is obsessed with subtitles. 2022, Preply.

Forsberg, C. Subtitles become popular among general population due to changes in TV, film consumption. ABC News: Australian Broadcasting Corporation., 2023.

Cunningham, K., Mumbling actors, bad speakers or lazy listeners? Why everyone is watching TV with subtitles on, in The Guardian. 2023.

McCue, T., Verizon Media Says 69 Percent Of Consumers Watching Video With Sound Off, in Forbes. 2019.

Having conducted a study on two continents among a large number of participants, and having analysed the data with inferential statistics, whose role is to allow us to extrapolate our findings to other populations, extending beyond our research sample, we believe that the results are of interest to the community - both and the general public and researchers, particularly from the fields such as Translation Studies, Audiovisual Translation, Applied Linguistics, Psycholinguistics, Cognitive Psychology and Psycholinguistics. 

- Interpretations of the results should be provided in light of previous studies and training theoriesز

Thank you for this comment. We have added more studies and theories to the introductory part of the manuscript, in particular Schnotz (2005) and Mayer (2005 and 2014). Please see the next point for details.

- No relevant theories were included in the discussion to support or provide anchorage to the research problem.

We added a section “Theoretical framework” in the introductory part of the manuscript. We hope it provides support to the research problem presented. We also added references to the theories in the Discussion.

Theoretical framework

Watching subtitled video requires the coordination of complex mental processes that have not been explored extensively in the literature. There are some theoretical models and frameworks to explain the integration of text, image and sound, such as the integrated model of text and picture comprehension [16]; the multimodal integrated-language framework [17, 18]; and the cognitive theory of multimedia learning [19, 20]. The framework by Liao at al. [17] investigates this integration in the context of dynamic texts like video with subtitles and a soundtrack. Both the model by Schnotz [16] and Mayer’s theory [20] understand the cognitive architecture as based on input from auditory and visual sensory modalities and a limited working memory capacity. They highlight how split attention in contexts like watching subtitled video without sound could impede comprehension. In Schnotz’s model [16], split attention would result in both descriptive processing (of words) and depictive processing (of sounds and images) being dependent on visual perception, thereby increasing cognitive load. In Mayer’s theory [20], the multimedia principle holds that a combination of words and pictures are better for learning than words alone, whereas the modality principle assumes that presenting information in two modalities is better for comprehension than presenting information only in one modality. This theory, like Schnotz’s model, would imply that subtitled video without sound would therefore increase cognitive load.

Neither Schnotz’s model nor Mayer’s theory, however, specify how the processing of one source of information might impact on another (e.g., video and subtitles). The multimodal integrated-language framework of Liao et al. [17, 18] builds on models of eye movement control during reading to predict how the reading of subtitles might be impacted by the concurrent presence of visual and auditory information. The framework assumes that reading is a serial process (i.e., sentence processing is contingent on the strictly sequential identification of words) and that objects in a scene likewise have to be fixated sequentially for identification due to the limitations of visual perception. However, the auditory processing of speech and sounds and the tracking of previously identified objects can occur in parallel with these serial processes. In the absence of sound, the benefit of parallel processing of sound disappears, forcing the viewer to rely on the sequential processing of image and subtitles.

This article will attempt to confirm some of the findings of Liao et al. [17], by exploring whether greater reliance on subtitles in the absence of sound will result in more time on the subtitles, longer fixations, and shorter saccades (or lower word skipping), reflecting the prioritization of the subtitles when the benefit of parallel processing of sound is absent.

Many thanks to the study team for the remarkable research they conducted and the unique manner in which it was presented.

Thank you! We appreciate your positive feedback on our work.

Reviewer 2

1.The paper frequently employs long sentences, and breaking some of them would improve overall comprehension.

Thank you for this comment. Following your suggestion, we have revised the manuscript and shortened long sentences to improve readability.

2.Elaborate on how the sample size was estimated (justification).

When designing the study and calculating the required minimum number of participants, we followed the suggestions put forward by Brysbaert (2019):

- We used a within-subject (rather than between-subject) design to increase power.

- In the eye-tracking part, we used multiple observations per condition per participant (each subtitle in each clip was used as a data point in each of the two conditions for each participant) in mixed models analyses. 

- For the ANOVA test, a minimum of 27 participants per condition in a two-way ANOVA with one repeated-measures factor is recommended, assuming the power of 80% and a medium effect size.

- We aimed to recruit at least 40 participants per group on the assumption that some data will have to be discarded due to poor quality, as is normally the case in any eye-tracking study.

Source:

Brysbaert, M. (2019). How Many Participants Do We Have to Include in Properly Powered Experiments? A Tutorial of Power Analysis with Reference Tables. Journal of Cognition, 2(1), 16. https://doi.org/10.5334/joc.72

3.Sampling technique employed (how) and related sampling bias should be discussed.

We used convenience and snowball sampling methods. Participants were recruited based on the criteria of being either a Polish or English native speaker, between the ages of 18 and 55. We used the pools of participants available at the institutions taking part in the study, social media, notice boards, and personal contacts to recruit participants. 

At Macquarie University, we could only use people on campus, and as the campus is rather distant from the city, we effectively tested students. This resulted in a skewed sample of Australian participants, and a resulting confound with age: as we acknowledge in the manuscript, age was confounded with the participants provenance.

4.The justification of why different apparatuses were used for eye movement measurements, as well as differing distances from the screen and potential biases, should be elaborated on.

Many thanks for asking an important question. We used an EyeLink Plus 1000 (University of Warsaw, Poland; Macquarie University, Sydney, Australia) and a Portable Duo (University of East Anglia, UK) because these are the systems the research group had in place in the different locations of data collection. Prior to data collection, we enquired precisely about this with the manufacturer (SR Research, Canada), who confirmed that they have done extensive testing to determine the specifications of both systems, and the results show that data quality is comparable for the two systems.

The technical characteristics of both systems are provided below.

Eyelink 1000 Plus: https://www.sr-research.com/wp-content/uploads/2017/11/eyelink-1000-plus-specifications.pdf

Eyelink Portable Duo: https://www.sr-research.com/wp-content/uploads/2017/07/portable-duo-specifications.pdf

First of all, the technical characteristics of both systems are similar and fully comparable. Second, the data file format is the same across the two systems. Third, the systems use the same underlying technology, including the same eye tracking algorithms. Fourth, experiment creation procedures (through the Experiment Builder software) and data analysis options (in the Data Viewer software) are the same. We ran exactly the same Experiment Builder project and used exactly the same data analysis scripts for both setups.

The differences across the systems (e.g., cable types, mounting options) should not affect data quality, data collection, or data analysis pipelines. The reviewers can find a side-by-side comparison of specs from the manufacturer here, where these differences are described: https://www.sr-research.com/wp-content/uploads/2021/03/EyeLink-Portable-Duo-and-1000-Plus-Specs.pdf

Key experimental elements that could introduce discrepancies and therefore confounds are (a) the sampling frequency used during data collection, (b) presentation rates of the videos, (c) refresh rates of the monitor screen and (d) the average accuracy of the machines. We made sure that (a) all data was recorded at 1000 Hz in both systems, (b) presentation rates were kept the same in all videos in both setups (25 fps, frames per second), (c) monitor refresh rates were the same (60 Hz) and (d) the accuracy specs of both systems is also the same (typically 0.25° to 0.5°, as you can see from the links above). 

Lastly, as for the different eye-to-screen distances, these are determined by the size and position of the monitor screen. A different distance is not synonymous with incompatibility but is in fact required for letters to subtend a comparable portion of the visual angle in both setups. Because the Portable Duo is mounted on a specialised laptop (provided by the manufacturer) with predetermined screen dimensions, whereas the 1000 Plus is used with a separate monitor screen, in order for the participants to see the same size subtitle letters, sitting at different distances is necessary. As we stated in the method section, in both setups each subtitle letter subtended ~0.32° of the visual angle, which ensured comparability and was achieved by finding the ideal distance between t

---

## [Decision Letter · Decision Letter 1]

14 Jun 2024

Watching subtitled videos with the sound off affects viewers’ comprehension, cognitive load, immersion, enjoyment, and gaze patterns: A mixed methods eye-tracking study

PONE-D-23-41429R1

Dear Dr. Szarkowska,

We’re pleased to inform you that your manuscript has been judged scientifically suitable for publication and will be formally accepted for publication once it meets all outstanding technical requirements.

Kind regards,

Antonio Peña-Fernández, PhD

Academic Editor

PLOS ONE

Additional Editor Comments (optional):

Dear authors,

Thank you for addressing the comments provided by the reviewers. I recommend its publication in our journal, congratulations.

Best wishes,

Antonio

Reviewers' comments:

Reviewer's Responses to Questions

**Comments to the Author**

1. If the authors have adequately addressed your comments raised in a previous round of review and you feel that this manuscript is now acceptable for publication, you may indicate that here to bypass the “Comments to the Author” section, enter your conflict of interest statement in the “Confidential to Editor” section, and submit your "Accept" recommendation.

Reviewer #3: All comments have been addressed

2. Is the manuscript technically sound, and do the data support the conclusions?

Reviewer #3: Yes

3. Has the statistical analysis been performed appropriately and rigorously? 

Reviewer #3: Yes

4. Have the authors made all data underlying the findings in their manuscript fully available?

Reviewer #3: Yes

5. Is the manuscript presented in an intelligible fashion and written in standard English?

Reviewer #3: Yes

6. Review Comments to the Author

Reviewer #3: Overall it is good effort of author(s). The author responded all queries from the reviewer. The author mentioned the sampling techniques with proper referencing. Mixed method triangulation was also properly defined now. The author(s) did a good comparison between two languages.

This article will be a good contribution in domain of knowledge. It is recommended.

7. PLOS authors have the option to publish the peer review history of their article (what does this mean?). If published, this will include your full peer review and any attached files.

Reviewer #3: No

---

## [Editor Report · Acceptance letter]

2 Jul 2024

PONE-D-23-41429R1 

PLOS ONE

Dear Dr. Szarkowska, 

I'm pleased to inform you that your manuscript has been deemed suitable for publication in PLOS ONE. Congratulations! Your manuscript is now being handed over to our production team.

Kind regards, 

on behalf of

Dr. Antonio Peña-Fernández 

Academic Editor

PLOS ONE